# 8th Edition Tumor, Node, and Metastasis T-Stage Prognosis Discrepancies: Solid Component Diameter Predicts Prognosis Better than Invasive Component Diameter

**DOI:** 10.3390/cancers12061577

**Published:** 2020-06-15

**Authors:** Kazuhito Funai, Akikazu Kawase, Kiyomichi Mizuno, Shin Koyama, Norihiko Shiiya

**Affiliations:** First Department of Surgery, Hamamatsu University School of Medicine, Hamamatsu 431-3192, Japan; akawase@hama-med.ac.jp (A.K.); k.mizuno@hama-med.ac.jp (K.M.); skoyama@hama-med.ac.jp (S.K.); shiyanor@hama-med.ac.jp (N.S.)

**Keywords:** lung cancer prognosis, pathological invasive size, solid part, T descriptor, TNM classification

## Abstract

The biggest change in the 8th edition of the tumor, lymph node, and metastasis (TNM) classification is the recommendation of the solid component diameter and invasive size for determining the clinical and pathological T-factor, respectively. Here, we validated new proposals for the Lung Cancer TNM classification’s revision and compared clinical and pathological T-stages. We retrospectively analyzed 177 cases of non-small cell lung cancers without lymph node metastasis, and involving complete resection, that occurred in our department between January 2017 and March 2019. We reviewed the overall tumor diameter, solid component diameter, and clinical T-factor on computed tomography (CT), and the pathological tumor diameter, pathological invasion diameter, pathological T-factor, and prognosis. The difference between the pathological invasive size and solid size on CT was within 5 mm in 99 cases (56%). At a two-year recurrence-free survival rate, the clinical T-stage demonstrated a better prognostic outcome than the pathological T-stage. Despite including the benign findings, the solid component diameter was better correlated with prognosis than the invasive size. Therefore, in cases of discrepancies of clinically and pathologically detected tumor size, the solid CT size should also be used for the pathological T classification.

## 1. Introduction

T-descriptors were announced by the International Association for the Study of Lung Cancer (IASLC) in the tumor, node, and metastasis (TNM) classification’s 8th edition for lung cancer in 2015 [1]. The biggest change in this edition is the recommendation of the solid and the invasive component diameter for determining the clinical and pathological T-factor, respectively. Particularly, for part-solid tumors, the invasive component’s size is used to assign the T category [2]. However, the invasion size was incorporated into lung cancer’s T-factor classification from a literature search without any large-scale investigation [2].

For some time, there have been disagreements among pathologists regarding the diagnosis of lung adenocarcinoma subtypes [3]. When lung adenocarcinoma diagnoses individually reviewed by two lung pathologists are compared, the diagnostic concordance is only 82.4% [4]. Moreover, the measurement of the pathological invasion diameter differs among pathologists who specialize in lung cancer. Urer et al. reported that the agreement between deciding whether the case is minimally invasive or invasive was low [3]. General hospitals, more so than high-volume centers dedicated to cancer, often question the accuracy of the diameter of pathological invasion measurements in signifying invasiveness.

Therefore, we aimed to clarify which best reflects prognosis, the clinical T-factor or the pathological T-factor, on computed tomography (CT) using cases that underwent lobectomy, and examined whether the pathological T-factor diagnosed in actual medical practice is accurate.

## 2. Result

Clinicopathological characteristics of the patients are shown in Table 1. The histological types among the 177 cases included 125 adenocarcinomas, 34 squamous cell carcinomas, 7 large cell neuroendocrine carcinomas (LCNEC), and 11 other histological types. The median follow-up was 624 days (range, 97–1207 days), and the median time from the CT scan to surgery was 6 days (range, 1–60 days). The median size of the overall tumor diameter was 22 mm (range, 6–94 mm), the solid component diameter was 20 mm (range, 0–94 mm), the pathological tumor diameter was 25 mm (range, 8–98 mm), and the pathological invasive diameter was 18 mm (range, 0–98 mm). The difference between the pathological and the overall tumor diameter at CT was a median of −1 mm (−28 to +41 mm). The difference between the pathological invasive and the solid component diameter on CT was −1.4 mm (−34 to +97 mm), and the difference between the two was within 5 mm in 99 cases (56%). The distribution of cases by stage is shown in Table 2. In many cases of T1b and T1c, the pathological T-stage was lower than the clinical T-stage (Table 3). Conversely, most cases with staging from T1 to T2 or higher were pleural invasion (pl) or pulmonary metastasis (pm). Fourteen cases were diagnosed as pathological T3, of which eight were pl, three were size criteria, and three were pm. Four cases diagnosed as pathological T4, two in size criteria, and two in mediastinal invasion (Table 3). Of 18 cases of pT3 and pT4, 8 cases were adenocarcinoma and 7 were squamous cell carcinoma. Twenty-eight patients received postoperative adjuvant chemotherapy. Only five patients received platinum-based adjuvant chemotherapy: two patients with LCNEC, two patients with T4N0M0, and one patient who participated in the clinical trial. Tegafur-uracil was administered to 23 adenocarcinomas of 2 cm or larger.

The two-year recurrence-free survival (RFS) rates for each clinical T-stage were 100%, 100%, 100%, 93.7%, 92.0%, 75.2%, 100%, 50%, and 100% for Tis, T1mi, T1a, T1b, T1c, T2a, T2b, T3, and T4, respectively, and for each pathological T-stage were 100%, 95.0%, 92.9%, 95.9%, 100%, 74.1%, 50%, 100%, and 100% for Tis, T1mi, T1a, T1b, T1c, T2a, T2b, T3, and T4, respectively (Table 4).

The patient characteristics of 125 adenocarcinomas are shown in Table 5. There were 66 females and 59 males. The median age was 70 years (range 41–87). In total, 56 cases were over 70 years old, 80 cases were heavy smokers (Brinkman index > 400), and 115 cases had a normal percent predicted forced expiratory volume in one second (%FEV1.0). The histological subtype of the adenocarcinomas included 51 lepidic adenocarcinomas, 45 papillary adenocarcinomas, 12 acinar adenocarcinomas, 2 micropapillary adenocarcinomas, 2 solid adenocarcinomas, 1 acinar and solid adenocarcinoma (35% each), and 12 variants of invasive adenocarcinomas. Variants of invasive adenocarcinomas included 10 invasive mucinous adenocarcinomas and 2 colloid adenocarcinomas (Table 5). The high carcinoembryonic antigen (CEA) level was observed in 28 cases (36%) pleural invasion in 22 (18%), lung metastasis in 2 (2%), blood vessel invasion in 44 (35%), lymphatic vessel invasion in 30 (24%), and spread through alveolar space in 35 cases (28%) (Table 5).

Among 125 adenocarcinoma cases, overall tumor size was almost the same compared with all histological types; however, solid component, pathological tumor, and invasive size were smaller than all histological types. Moreover, the agreement rate of clinical and pathological T-size was low (52%) (Table 6). Pathologically, 4 cases from T1a, 12 cases from T1b, and 1 case from T1c were diagnosed as minimally invasive adenocarcinoma (T1mi) (Table 7).

The two-year RFS rates for each clinical T-stage (Figure 1) were 100%, 100%, 100%, 97.8%, 100%, 77.8%, 66.7%, and 100% for Tis, T1mi, T1a, T1b, T1c, T2a, T3, and T4, respectively, and for each pathological T-stage (Figure 2) were 100%, 100%, 91.7%, 100%, 100%, 84.1%, 100%, 100%, and 100% for Tis, T1mi, T1a, T1b, T1c, T2a, T2b, T3, and T4, respectively.

## 3. Discussion

New entities of adenocarcinoma in situ (AIS), minimally invasive adenocarcinoma (MIA), and lepidic predominant adenocarcinoma were defined in 2011 and were adopted into the 2015 WHO lung cancer classification [5]. In the 8th edition of the TNM classification, these entities were adapted to the T component staging system [6]. AIS was classified as Tis and MIA as T1mi. However, before this new classification system, 33,115 of the 70,967 non-small cell lung cancers (NSCLC) collected between 1999 and 2010 were analyzed, and Rami-Porta et al. recommended that T-stage be classified by tumor size as follows: “to subclassify T1 into T1a (≤1 cm), T1b (>1 to ≤2 cm), and T1c (>2 to ≤3 cm); to subclassify T2 into T2a (>3 to ≤4 cm) and T2b (>4 to ≤5 cm); to reclassify tumors greater than 5 cm to less than or equal to 7 cm as T3; and to reclassify tumors greater than 7 cm as T4 [1]”. This report did not provide descriptions of the invasive part. However, Travis et al. reviewed a great deal of literature and proposed that for non-mucinous adenocarcinoma of the lung, the pathological tumor size be determined according to the invasive size excluding lepidic components, and clinical tumor size be determined by the invasive component size excluding ground glass [2]. Despite the disadvantage of this reclassification having no basis in large data studies, the Union for International Cancer Control’s (UICC) recommendation to measure tumor size by invasive size was first adopted in the TNM classification of lung cancer [6]. It was groundbreaking and introduced fundamental changes; however, these changes left future challenges. Although two clinical studies were reported from Japan and Korea to validate the new eighth edition of the TNM classification [7,8], the invasion size being incorporated into the T-factor classification of lung cancer from a literature search without a large-scale investigation remains a problem [2]. In radiation oncology, the presence of a ground glass opacity (GGO) component has been confirmed to impact the prognosis of patients with surgically resected NSCLC [9,10,11], and a solid size on thin-section CT image indicates a better prognosis than a classic tumor size [12]. Additionally, in sublobar resection for stage IA lung adenocarcinoma, the T descriptor, represented by the solid component size rather than total tumor size, was a better predictor of recurrence [13]. The radiological solid component size corresponds well with pathologic invasiveness of lung adenocarcinoma [7,14]; however, Yanagawa et al. reported that the CT-measured invasive size was larger than that of the actual invasive component [14]. In other words, while everyone is aware of the discrepancy between clinical invasive size and pathological invasive size, no studies have compared which T-factor correlates with prognosis after lobectomy.

Travis et al. provided a list of key questions to guide future research in assessing tumor size by TNM classification in lung adenocarcinomas presenting with subsolid nodules by CT or with a lepidic component by pathologic examination. We have considered this list of key questions and in particular the following from a practical perspective: “What is the reproducibility of measuring size in invasive versus lepidic components, and how can this be improved [2]?”

In the current study, the coincidence rate between the solid component diameter on CT and the pathological invasive diameter was low, and the clinical T-stage demonstrated a superior prognostic outcome compared with the pathological T-stage. Specifically, with pathological T-factors, the two-year RFS rate of T1mi and T1a, which should have a good prognosis, is low (Table 4). This indicates that the analysis of pathological T-factor in our institution is inaccurate. However, this problem is not only in our facility but also in pathological diagnostic centers worldwide. This is because the concordance rate for the pathological diagnosis of lung adenocarcinoma, especially for invasion, is known to be low, even among pulmonary pathologists [3,4]. Six pulmonary pathologists digitally reviewed 60 slides of small lung adenocarcinoma in three rounds, and interobserver agreement was fair to moderate; the range of the measured invasive component in a single case was up to 19.2 mm among observers [15]. Generally speaking, the prognostic significance of pathological invasive size and solid component diameter is thought to be important for adenocarcinoma with GGO, but it is not the only adenocarcinoma. In interstitial pneumonia-based squamous cell carcinoma, the pathological tumor size is often larger than the CT tumor size. This is because the cancer cells invade the honeycomb lung, which is caused by interstitial pneumonia around the tumor, and the boundary of the tumor cannot be seen on the CT image. In fact, discrepancies of up to 97 mm between solid components found in the radiological and pathological assessments in this study were such cases. In addition, the cases with extremely large differences were such squamous cell carcinomas. Therefore, not only adenocarcinoma but also other NSCLC histological type were included in this study. The difference became even greater when only adenocarcinoma was extracted and examined. Pathological T-stage should have a more accurate prognosis than clinical T-stage in patients who are up stage from T1 due to pleural invasion or micropulmonary metastasis. Despite these cases, the prognosis of pathological T is worse than that of clinical T, especially in T1, indicating the difficulty in measuring the pathological invasion size.

Here, 28 patients received postoperative adjuvant chemotherapy. There is no evidence of platinum-based adjuvant chemotherapy in this study cohort because all cases were N0 and stage I, according to TNM classification at the time of LACE trial [16]. Therefore, only five patients received platinum-based adjuvant chemotherapy: two patients with LCNEC, two patients with T4N0M0, and one patient who participated in the clinical trial. Tegafur-uracil was administered to adenocarcinomas of 2 cm or larger with consent, according to the lung cancer clinical practice guidelines of the Japan Lung Cancer Society [17,18].

This study has some limitations. First, it is a small retrospective study of a single institute, and the median follow-up was short (624 days). However, even with a short median follow-up time, the clinical T-stage in T1mi and T1a clearly reflects that the prognosis was better than the pathological T-stage in the RFS period. The early recurrence of T1mi, which should cause almost no recurrence, and T1A, which has a good prognosis, indicates that pathological invasion is difficult to measure. There is also a bias in published papers. Many papers are rejected because of the small number of cases, and the valuable truth may have been buried. Second, there is a need for future validation studies. Although this study needs to be re-validated using a large number of cases, it is valid as it points out potential problems with the 8th edition of the TNM classification. In a previous study, pathological results were reevaluated and invasive size was remeasured for the study [7,8,10,13]. These papers have been published by national cancer centers or high-volume centers in countries where there are pathologists specializing in lung cancer. It is not surprising that the invasion data remeasured for research by lung cancer pathologists who are familiar with lung cancer diagnosis correlates with prognosis. However, we think that this is far from the current state of pathological diagnosis performed worldwide. Unbiased data of larger-cohort studies performed worldwide are necessary to examine clinical T factors and pathological T factors used in clinical practice. For example, in Japan, the Japan Lung Cancer Society, the Japanese Association for Chest Surgery, and the Japanese Respiratory Society jointly established the Japanese Joint Committee for Lung Cancer Registration, which has regularly maintained lung cancer registries for surgical cases stratified as five-year periods. They analyzed data obtained in these registries to reveal the most recent surgical outcomes [19]. Similar efforts should be made at the IASLC.

Aside from pathologists who specialize in lung cancer and work in high-volume centers, it is difficult to correctly diagnose pathological T-factors by pathological invasive size, especially in emerging and developing countries. TNM classification of lung cancer should be consistent worldwide, even in emerging and developing countries. Strictly speaking, the solid component size upon CT is not the same as the invasive size in pathology. Moreover, the presence of active fibroblastic proliferation and the size of central fibrosis in a tumor have been known as bad prognostic factors [20,21], because the solid components on CT include benign findings such as structural collapse of the alveoli [20] and do not reflect only the invasion of cancer. However, even with the inclusion of those benign findings, in this study, the solid component diameter on CT better correlated with the prognosis than the invasive component diameter on pathology. Using the pathological invasive diameter in all hospitals is inappropriate because it is inferior to the solid component diameter as a prognostic factor. Therefore, in extreme cases, many general hospitals may be able to replace the pathological invasive size with the solid CT size after histological diagnosis and pathological diagnosis of pleural or chest wall invasion. This is not an impractical idea and is actually stated in the UICC TNM supplement that “in cases of discrepancies of clinically and pathologically detected tumor size the clinical measurement should be used also for the pathological T classification” [22].

## 4. Materials and Method

### 4.1. Patients Selection

We retrospectively reviewed the medical records of the 201 consecutive patients with non-small cell lung cancers (NSCLC) that underwent complete resection at our department between January 2017 and April 2019, and 177 patients without lymph node metastasis were included in the study. All patients underwent complete resection of the tumor and hilar and mediastinal lymph nodes. Patients who received induction chemotherapy or radiotherapy, and with evidence of a residual tumor at the surgical margin, were excluded from this study. The Ethics Committees of the Hamamatsu University School of Medicine (approval number 17–157) approved the protocol. We retrospectively reviewed the overall tumor diameter, solid component diameter, and clinical T-factor on CT, as well as the pathological tumor diameter, pathological invasion diameter, pathological T-factor, and prognosis.

### 4.2. Pathological Evaluation

Surgical specimens were fixed with 10% formalin and embedded in paraffin. Serial 4-μm sections were stained with hematoxylin and eosin. The sections stained by the Elastica Van Gieson method were examined to measure the invasive size and pleural invasion. For the pathological measurement of the tumor size, first, pathological tumor size was grossly measured to estimate three-dimensional sizes. Next, the tumor size was reevaluated at the time of microscopic evaluation of the tumor. Invasive size was confirmed on glass slides for all lesions. For tumors larger than could fit on a single glass slide, complete tumor cross sections were taken from each tumor that could be reconstructed over multiple glass slides. We have not used the calculated invasive tumor size [23], because all lepidic is measured as non-invasive by that method. The cases were pathologically staged based on the 8th edition of the TNM classification for lung cancer [6]. Histopathological studies were performed according to the World Health Organization (WHO) criteria [5].

The pathologists were unaware of the radiologic appearance and features before grossing. Two pathologists double-checked the pathological diagnosis. Eleven pathologists prepared the first report, and two senior pathologists performed the final check. Pathological factors were picked up from the pathological report created in this way.

### 4.3. Radiological Evaluation

The solid diameter was measured using thin-section CT images at 1-mm collimation or a 0.1 mm thick image of SYNAPSE VINCENT (FUJIFILM Corporation, Tokyo, Japan). The radiologic measurement of the tumor size, the single largest dimension measured, was performed using electronic calipers on thin-sections CT. For the maximum diameter, coronal and sagittal section measurements were used in addition to the CT cross section. Thoracic surgeons remeasured the tumor and invasion size at the pre-operative conference with reference to the CT report double-checked by two radiologists. We measured the maximum diameter of the tumor including a ground glass opacity (GGO) and a solid part only in the lung window.

### 4.4. Patient Follow-up

Patients were evaluated at 3-month intervals. The follow-up evaluation included physical examination, chest radiography, and blood testing including pertinent tumor markers. Thoracoabdominal CT was performed at least once a year, and whenever any symptoms or signs of recurrence were detected, further evaluations were conducted, including chest and abdominal CT, brain magnetic resonance imaging, bone scintigraphy, and positron emission tomography/CT. Recurrence was diagnosed based on compatible physical examination and diagnostic imaging findings, and the diagnosis was histologically confirmed when clinically feasible.

### 4.5. Statistical Analysis

Survival was estimated using the Kaplan-Meier method, and any differences in survival were determined using the log-rank analysis. Recurrence-free survival (RFS) was the time from curative surgery to recurrence or death. Patients without recurrent disease were censored at the last time they were known to be recurrence-free. Two-sided *p* values ≤ 0.05 were considered statistically significant. All statistical analyses were performed with the SPSS statistics software package version 25 (IBM SPSS statistics).

## 5. Conclusions

The coincidence rate between the solid component diameter on CT and the pathological invasive diameter was low, and the clinical T-stage demonstrated a superior prognostic outcome compared with the pathological T-stage. Therefore, in cases of discrepancies of clinically and pathologically detected tumor size, the solid CT size should also be used for the pathological T classification.

## Figures and Tables

**Figure 1 cancers-12-01577-f001:**
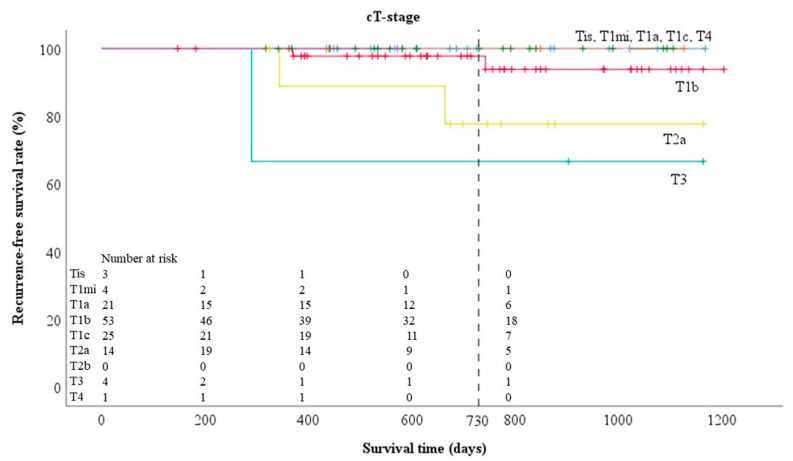
Prognosis by clinical T-factor in 125 adenocarcinomas.

**Figure 2 cancers-12-01577-f002:**
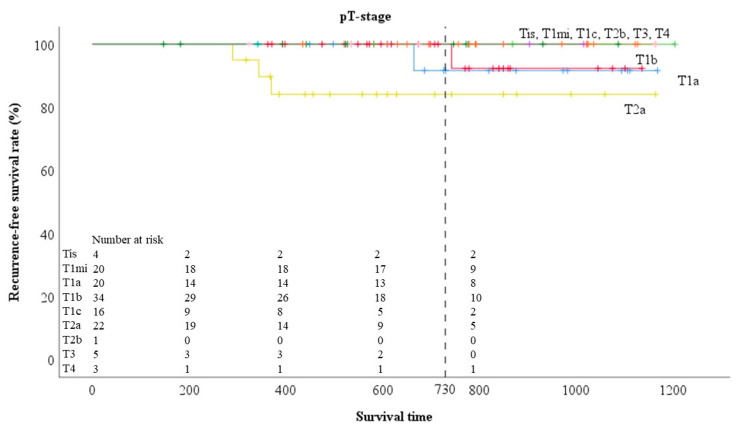
Prognosis by pathological T-factor in 125 adenocarcinomas.

**Table 1 cancers-12-01577-t001:** Clinicopathological characteristics of patients.

Characteristics	*n*
Histological types	
Adenocarcinoma	125 (71%)
Squamous cell carcinoma	34 (19%)
Large cell neuroendocrine carcinoma	7 (4%)
Others	11 (6%)
Median follow up (days)	624 (range, 97–1207)
Median time from CT scan to surgery (days)	6 (range, 1–60)
Number of diagnostic pathologists	11 people
Overall tumor size (T)	22 mm (range, 6–94)
Solid component size (Ts)	20 mm (range, 0–94)
Pathological tumor size (P)	25 mm (range, 8–98)
Pathological invasive size (Pi)	18 mm (range, 0–98)
P minus T	−1 mm (−28 to +41)
Pi minus Ts	−1.4 mm (−24 to +97)
−5 mm < Pi minus Ts < +5 mm	99 (56%)

**Table 2 cancers-12-01577-t002:** Distribution of cases by T-stage.

T-Stage	Clinical T-Stage (People)	Pathological T-Stage (People)
Tis	3 (2%)	4 (2%)
T1mi	4 (2%)	20 (11%)
T1a	23 (13%)	22 (13%)
T1b	72 (41%)	51 (29%)
T1c	38 (21%)	25 (14%)
T2a	21 (12%)	31 (18%)
T2b	1 (1%)	6 (3%)
T3	9 (5%)	14 (8%)
T4	6 (3%)	4 (2%)

**Table 3 cancers-12-01577-t003:** Number of cases changed from clinical T-stage to pathological T-stage.

Clinical T-Stage		Pathological T-Stage	
cT-stage	*n*	Changed from cT-stage to pT-stage	*n*
Tis	3	No change: 33%	Tis: 1
		Upstage: 66%	T1a: 1 T1b: 1
T1mi	4	No change: 75%	T1mi: 3
		Upstage: 25%	T1a: 1
T1a	23	Downstage: 26%	Tis: 2 T1mi: 4
		No change: 35%	T1a: 8
		Upstage: 39%	T1b: 7 T1c: 1 T4: 1
T1b	72	Downstage: 28%	Tis: 1 T1mi: 12 T1a: 7
		No change: 51%	T1b: 37
		Upstage: 21%	T1c: 4 T2a: 10 (pl) T3: 1 (pm)
T1c	38	Downstage: 24%	T1mi: 1 T1a: 4 T1b: 4
		No change: 34%	T1c: 13
		Upstage: 42%	T2a: 10 (pl: 9) T2b: 2 T3: 4 (pl3: 2, pm: 2)
T2a	21	Downstage: 43%	T1a: 1 T1b: 2 T1c: 6
		No change: 38%	T2a: 8
		Upstage: 19%	T2b: 1 T3: 3
T2b	1	No change: 100%	T2b: 1
T3	9	Downstage: 45%	T2a: 3 T2b: 1
		No change: 45%	T3: 4
		Upstage: 10%	T4: 1
T4	6	Downstage: 67%	T1c: 1 T2b: 1 T3: 2
		No change: 33%	T4: 2

**Table 4 cancers-12-01577-t004:** Two-year recurrence-free survival rates for each T-stage.

T-Stage	Clinical T-Stage	Pathological T-Stage
Tis	100%	100%
T1mi	100%	95.0%
T1a	100%	92.9%
T1b	93.7%	95.9%
T1c	92.0%	100%
T2a	75.2%	74.1%
T2b	100%	50%
T3	50%	100%
T4	100%	100%

**Table 5 cancers-12-01577-t005:** Clinicopathological characteristics of patients in 125 adenocarcinomas.

Characteristics	*n*
Adenocarcinoma	
Lepidic	51 (40%)
Acinar	12 (9%)
Papillary	45 (36%)
Micropapillary	2 (2%)
Solid	2 (2%)
Acinar and solid (35% each)	1 (1%)
Variants (Invasive mucinous)	10 (8%)
(Colloid)	2 (2%)
Age	
<70	69 (55%)
≥70	56 (45%)
Sex	
Female	66 (53%)
Male	59 (47%)
CEA	
≤4.4	97 (78%)
>4.5	28 (22%)
Brinkman index	
≤400	80 (64%)
>400	45 (36%)
%FEV1.0	
<70%	119 (95%)
≥70%	6 (5%)
Pleural invasion	
Positive	22 (18%)
Negative	103 (82%)
Pulmonary metastasis	
Positive	2 (2%)
Negative	123 (98%)
Lymphatic vessel invasion	
Positive	30 (24%)
Negative	95 (76%)
Blood vessel invasion	
Positive	44 (35%)
Negative	81 (65%)
Spread through alveolar space	
Positive	35 (28%)
Negative	73 (58%)
Unknown	17 (14%)

**Table 6 cancers-12-01577-t006:** Comparison of measurement diameter between adenocarcinoma and all histological types.

Characteristics	All Histlogical Types	Adenocarcinoma
Median follow up (days)	624 (range, 97–1207)	651 (range, 97–1207)
Median time from CT scan to surgery (days)	6 (range, 1–60)	6 (range, 1–60)
Overall tumor size (T)	22 mm (range, 6–94)	22 mm (range, 8–64)
Solid component size (Ts)	20 mm (range, 0–94)	17 mm (range, 0–64)
Pathological tumor size (P)	25 mm (range, 8–98)	22 mm (range, 8–98)
Pathological invasive size (Pi)	18 mm (range, 0–98)	14 mm (range, 0–98)
P minus T	−1 mm (−28 to +41)	0 mm (−16 to +41)
Pi minus Ts	−1.4 mm (−24 to +97)	−1 mm (−32 to +97)
−5 mm < Pi minus Ts < +5 mm	99 (56%)	65 (52%)

**Table 7 cancers-12-01577-t007:** Number of cases changed from clinical T-stage to pathological T-stage in 125 adenocarcinomas.

Clinical T-Stage		Pathological T-Stage	
cT-stage	*n*	Changed from cT-stage to pT-stage	*n*
Tis	3	No change: 33%	Tis: 1
		Upstage: 66%	T1a: 1
			T1b: 1
T1mi	4	No change: 75%	T1mi: 3
		Upstage: 25%	T1a: 1
T1a	21	Downstage: 29%	Tis: 2
			T1mi: 4
		No change: 38%	T1a: 8
		Upstage: 33%	T1b: 5
			T1c: 1
			T4: 1 (size)
T1b	53	Downstage: 36%	Tis: 1
			T1mi: 12
			T1a: 6
		No change: 41%	T1b: 22
		Upstage: 23%	T1c: 4
			T2a: 8 (pl:8)
T1c	25	Downstage: 32%	T1mi: 1
			T1a: 3
			T1b: 4
		No change: 28%	T1c: 7
		Upstage: 40%	T2a: 8 (pl:7))
			T2b: 1
			T3: 1 (pm)
T2a	14	Downstage: 50%	T1a: 1
			T1b: 2
			T1c: 4
		No change: 36%	T2a: 5 (pl:1)
		Upstage: 14%	T3: 2 (pl:1, size:1)
T2b	0		-
T3	4	Downstage: 25%	T2a: 1
		No change: 50%	T3: 2
		Upstage: 25%	T4: 1 (size)
T4	1	No change: 100%	T4: 1

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
