# Peer review of "8th Edition Tumor, Node, and Metastasis T-Stage Prognosis Discrepancies: Solid Component Diameter Predicts Prognosis Better than Invasive Component Diameter"

_cancers, 2020, doi:10.3390/cancers12061577_

Round 1

Reviewer 1 Report

The authors provide a well written and to the point manuscript with the aim to compare the relevance of the pathological tumour invasive margin (pathological T-stage) and the tumour solid size (C-stage) in curatively resected NSCLC on recurrence rates. Therefore, they retrospectively evaluated a series of 177 NSCLC (all histology’s) without N-disease / pre-operative systemic/radiotherapy and with a R0 resection. A pathological and radiological evaluation is performed to evaluate the pathological invasive margin and the radiological tumour size using 1mm slide CT-scans. They find a poor concordance rate for clinical and pathological tumour size, mainly due to downgrading of clinical T-stage T1bà pathological T1mi. And upgrading of clinical T1a-cà T2a mainly due to pleural invasion. With a median follow-up of 2 years a slight increase in percentage of recurrences are found in pathological T1mi/T1a compared to more recurrence in clinical T1b and T1c. The authors conclude that in cases where clinical T-stage is downgraded in the pathological T-stage due to tumour size, the clinical stage should be used for prognostic stratification. Although the included number of patients is limited, the manuscript does highlight an important clinical question.

Major comments:

  • As specified by the authors, the current study is rather small for adequate prognostic evaluation. Also, follow-up length is limited for recurrence of disease. Furthermore, multiple histological subtypes are included and as also specified by the authors there is a difference per histological subtype with regard to the evaluation of the invasive margin. I.e. in adenocarcinoma’s GGO are common and a T1mi is more often diagnosed compared to LCNEC/SqCC. An analysis specifically separating adenocarcinomas vs all others histology’s would be of interest. Furthermore, the authors should provide more detail on the histological subtyping of the included adenocarcinomas.
  • Whom did evaluate the radiological tumour size and was this reviewed for this manuscript by one or more radiologists? Was a contrast enhanced CT-scan used in all patients?
  • An evaluation including HR with reference of clinical and pathological T for recurrence of disease would be informative.

Minor comments:

  • Line 46 is there a reference for this statement?
  • Line 54-55: ‘The pathological diagnosis was double checked by two pathologists, and the primary diagnosis was performed by a group of 11 pathologists’ à methods section?
  • What was the day of last follow-up check for the cohort? This should be specified.
  • Table 1, please provide % of total for histology
  • Table 2, please provide % per table
  • Table 3 should provide complete data on all T-stages. Currently clinical T2a-T4 is not provided.
  • An overview of number patients remaining/time point below the X-axis for figure 2 would be informative.
  • What % of patients received adjuvant CTx? As patients with TNM II (>T2a) are eligible, this information should be provided.

Author Response

Major comments

As specified by the authors, the current study is rather small for adequate prognostic evaluation. Also, follow-up length is limited for recurrence of disease. Furthermore, multiple histological subtypes are included and as also specified by the authors there is a difference per histological subtype with regard to the evaluation of the invasive margin. I.e. in adenocarcinoma’s GGO are common and a T1mi is more often diagnosed compared to LCNEC/SqCC. An analysis specifically separating adenocarcinomas vs all others histology’s would be of interest. Furthermore, the authors should provide more detail on the histological subtyping of the included adenocarcinomas.

Whom did evaluate the radiological tumour size and was this reviewed for this manuscript by one or more radiologists? Was a contrast enhanced CT-scan used in all patients?

An evaluation including HR with reference of clinical and pathological T for recurrence of disease would be informative.

Response 1: We added analysis for 125 adenocarcinoma cases (Result; Line 80-102, Table 5-7). And we also added histological subtyping of the included adenocarcinomas to table5. Thoracic surgeons re-measured the tumor and invasion size at the pre-operative conference with reference to the CT report double checked by two radiologists (Line 253-255).

Contrast-enhanced CT was taken unless there was contrast agent allergy or renal dysfunction.

Since there are few events, we studied RFS instead of overall survival. Therefore, we think there are few events and it is difficult to detect HR accurately.

Minor comments:

  • Line 46 is there a reference for this statement?

Response: This is a personal opinion and there are no references.

  • Line 54-55: ‘The pathological diagnosis was double checked by two pathologists, and the primary diagnosis was performed by a group of 11 pathologists’ à methods section?

Response: As you pointed out, we moved to methods (Line 244-247).

  • What was the day of last follow-up check for the cohort? This should be specified.
  • Response: The answers to your questions are detailed in “Recurrence-free survival (RFS) was the time from curative surgery to recurrence or death. Patients without recurrent disease were censored at the last time they were known to be recurrence-free.”(Line 267-269)

  • Table 1, please provide % of total for histology
  • Table 2, please provide % per table

Response: As you pointed out, we added % to Figures 1 and 2.

  • Table 3 should provide complete data on all T-stages. Currently clinical T2a-T4 is not provided.

Response: As you pointed out, we added T2a-T4 data to table3.

  • An overview of number patients remaining/time point below the X-axis for figure 2 would be informative.

Response: We added number at risk to the figure.

  • What % of patients received adjuvant CTx? As patients with TNM II (>T2a) are eligible, this information should be provided.

Response: The post-operative adjuvant chemotherapy is described below (Result; Line 65-68, Discussion; Line 176-182).  However, there is no evidence of platinum-based adjuvant chemotherapy in this study cohort, because all cases were N0 and stage I according to TNM classification at the time of LACE trial

Reviewer 2 Report

In the manuscript entitled “8th edition Tumor, Node, and Metastasis T-stage prognosis discrepancies: solid component diameter predicts prognosis better than invasive component diameter” by Funai and colleagues, the authors compare the CT measurement of solid tumor component to the pathologic measurement of the invasive tumor size for a cohort of resected lung cancers, and compared these to recurrence-free survival rates. The authors show that clinical staging via CT is a better predictor of DFS than pathologic invasive tumor measurement in their hospital, and conclude that CT solid size should be used for pathologic T-classification.

Overall this is an interesting study, and one of interest to the lung cancer community as any updates to the forthcoming 9th edition AJCC tumor staging manual are considered. The study has merit; however, the authors should address the following to more clearly define the tumor cohort as well as add validity to the conclusions based on these findings:

1-The authors need to provide more granular details on how pathologic invasive size was actually measured/calculated. From the methods section “For the pathological measurement of the tumor size, first, pathological tumor size is taken by gross measurements to estimate three-dimensional sizes. Next, the tumor size was reevaluated at the time of microscopic evaluation of the tumor. Invasive size was confirmed on glass slides for all lesions.” How was invasive tumor size determined for tumors larger than could fit on a single glass slide for direct microscopic measurement (i.e. larger than ~1.5 cm)? Were complete tumor cross sections taken from each tumor that could be reconstructed over multiple glass slides? Was the invasive tumor size for lung adenocarcinomas calculated by multiplying the overall tumor size by the percentage invasive patterns (acinar+papillary+micropapillary+solid)? These details are very important, also see and cite/discuss in the discussion the following reference: “Anderson KR, Onken A, Heidinger BH, Chen Y, Bankier AA, VanderLaan PA. Pathologic T Descriptor of Nonmucinous Lung Adenocarcinomas Now Based on Invasive Tumor Size: How Should Pathologists Measure Invasion?. Am J Clin Pathol. 2018;150(6):499‐506. doi:10.1093/ajcp/aqy080  PMID: 30084917”

2-Invasive tumor size in the 8th edition AJCC largely refers only to non-mucinous lung adenocarcinomas: tumors that have a peripheral lepidic (i.e. inferred in-situ) component that correlates to the peripheral Ground Glass Opacity (GGO) seen on CT scan. Since this cohort represents a very heterogeneous group of lung tumors (adenocarcinomas and non-adenocarcinomas) that can have dramatically different clinical behaviors, why for this type of validation study don’t the authors only focus on adenocarcinomas to have a cleaner cohort that although will lower the overall “n” will likely provide cleaner results on which more definitive conclusions can be made?

3-For the CT overall tumor and solid component measurements, was the longest dimension taken from the same cut (axial/coronal/sagittal) or from which ever plane gave the largest measurement? Were all cuts reviewed? Why wasn’t the soft tissue window used to determine solid size (lung window is fine to measure total tumor size including the GGO component)?

4-It is a bit unusual that in Figure 2 the pathologic T3 and T4 tumors (n=18 if Table 2 is correct) all have 100% survival, and it appears as if this discrepancy is driving the discrepancies in outcome relative to the CT staging (Figure 1). How do the authors explain this? Were these large tumors overrepresented by a common type (i.e. squamous cell carcinoma or mucinous lung adenocarcinoma)? Was the upstaging of many cT1c tumors to pT2a/T3 on the basis of pleural invasion confirmed by elastic stains? How do the authors explain the cT1a that was pathologically upgraded to pT4? Was tumor spread through the airspaces (STAS) considered for the pathologic staging of tumors (exclusionary criteria for AIS and MIA)?   

5-This also brings up the point that the pathologic characterization of the tumor cohort is lacking. A more detailed pathologic breakdown of the cohort is needed. For the adenocarcinomas: how many were mucinous vs. non-mucinous, what was the predominant pattern (lepidic, acinar, solid, papillary, micropapillary), what was the status for tumor spread through the airspaces (STAS), etc?

Author Response

Point 1: The authors need to provide more granular details on how pathologic invasive size was actually measured/calculated. From the methods section “For the pathological measurement of the tumor size, first, pathological tumor size is taken by gross measurements to estimate three-dimensional sizes. Next, the tumor size was reevaluated at the time of microscopic evaluation of the tumor. Invasive size was confirmed on glass slides for all lesions.”

 How was invasive tumor size determined for tumors larger than could fit on a single glass slide for direct microscopic measurement (i.e. larger than ~1.5 cm)?

Were complete tumor cross sections taken from each tumor that could be reconstructed over multiple glass slides?

Was the invasive tumor size for lung adenocarcinomas calculated by multiplying the overall tumor size by the percentage invasive patterns (acinar+papillary+micropapillary+solid)?

These details are very important, also see and cite/discuss in the discussion the following reference: “Anderson KR, Onken A, Heidinger BH, Chen Y, Bankier AA, VanderLaan PA. Pathologic T Descriptor of Nonmucinous Lung Adenocarcinomas Now Based on Invasive Tumor Size: How Should Pathologists Measure Invasion?. Am J Clin Pathol. 2018;150(6):499506. doi:10.1093/ajcp/aqy080  PMID: 30084917

Response 1: For tumors larger than could fit on a single glass slide, complete tumor cross sections were taken from each tumor that could be reconstructed over multiple glass slides (Line 238-240).

We have not used the calculated invasive tumor size. Because all lepidic is measured as non-invasive by that method. (Line 240-241)

We also added reference 23.

Point 2: Invasive tumor size in the 8th edition AJCC largely refers only to non-mucinous lung adenocarcinomas: tumors that have a peripheral lepidic (i.e. inferred in-situ) component that correlates to the peripheral Ground Glass Opacity (GGO) seen on CT scan. Since this cohort represents a very heterogeneous group of lung tumors (adenocarcinomas and non-adenocarcinomas) that can have dramatically different clinical behaviors, why for this type of validation study don’t the authors only focus on adenocarcinomas to have a cleaner cohort that although will lower the overall “n” will likely provide cleaner results on which more definitive conclusions can be made?

Response 2: We also added analysis for 125 adenocarcinoma cases (Result; Line 80-102). And added Table 5-7, and replaced Figure 1.2 with adenocarcinoma cases.

Point 3: For the CT overall tumor and solid component measurements, was the longest dimension taken from the same cut (axial/coronal/sagittal) or from which ever plane gave the largest measurement? Were all cuts reviewed? Why wasn’t the soft tissue window used to determine solid size (lung window is fine to measure total tumor size including the GGO component)?

Response 3: The answers to the first two of your questions are detailed in “4.3. Radiological Evaluation” (lines 249–256). The coronal and sagittal view were also used to measure the tumor diameter.

According to the "Classification of Lung Cancer," the tumor diameter (total and solid size) is determined to measure lung window.

Point 4: It is a bit unusual that in Figure 2 the pathologic T3 and T4 tumors (n=18 if Table 2 is correct) all have 100% survival, and it appears as if this discrepancy is driving the discrepancies in outcome relative to the CT staging (Figure 1). How do the authors explain this? Were these large tumors overrepresented by a common type (i.e. squamous cell carcinoma or mucinous lung adenocarcinoma)? Was the upstaging of many cT1c tumors to pT2a/T3 on the basis of pleural invasion confirmed by elastic stains? How do the authors explain the cT1a that was pathologically upgraded to pT4? Was tumor spread through the airspaces (STAS) considered for the pathologic staging of tumors (exclusionary criteria for AIS and MIA)?  

Response 4: The details of Patholigical T3 and T4 are added to Line 61-64. Of 18 cases of pT3 and pT4, 8 cases were adenocarcinoma and 7 were squamous cell carcinoma (line 64-65). The sections stained by the Elastica Van Gieson method were examined for measuring the invasive size and pleural invasion (Line 234-235). There was no clear correlation between T3, T4 and STAS. Preoperative CT showed a nodule with GGO or honeycomb lungs.

In the pathology, cancer had invaded in those parts and the tumor size became large (Line 164-168).

Point 5: This also brings up the point that the pathologic characterization of the tumor cohort is lacking. A more detailed pathologic breakdown of the cohort is needed. For the adenocarcinomas: how many were mucinous vs. non-mucinous, what was the predominant pattern (lepidic, acinar, solid, papillary, micropapillary), what was the status for tumor spread through the airspaces (STAS), etc?

Response 5: We also added analysis for 125 adenocarcinoma cases (Result; Line 80-102, Table 5-7).

Round 2

Reviewer 1 Report

All questions have been sufficiently answered to provide readers more insight into the data presented. No further suggestions.

Author Response

Thank you very much for your constructive feedback on our manuscript.

We are grateful to you for their insightful suggestions and comments.

Reviewer 2 Report

The authors have significantly improved the overall quality of the manuscript by addressing most of the comments raised by the reviewers.

Because the data presented in Tables 3 and 7 is a bit dense (showing the breakdown of pathologic T-stages for each clinical/CT tumor stage group), I would suggest the authors identify some way to make it more clear when the change was an upstage, downstage, or no change in stage. This could be accomplished by color coding the text/background for each group respectively, or by separating with additional spaces/indents those that pathologically were upstaged vs. pathologically downstaged. Anything to make the format of these two tables more visually appealing and user-friendly.  

Author Response

Thank you very much for your constructive feedback on our manuscript. We are grateful to you for their insightful suggestions and comments.

We have added the changes in stage (upstage, downstage, or no change in stage) to Tables 3 and 7 and classified them by color.